# The (Re-)Emergence and Spread of Viral Zoonotic Disease: A Perfect Storm of Human Ingenuity and Stupidity

**DOI:** 10.3390/v15081638

**Published:** 2023-07-27

**Authors:** Veronna Marie, Michelle L. Gordon

**Affiliations:** 1Microbiology Laboratory, Department of Analytical Services, Rand Water, Vereeniging 1939, South Africa; 2School of Laboratory Medicine and Medical Sciences, College of Health Sciences, University of KwaZulu-Natal, Durban 4001, South Africa; tarinm@ukzn.ac.za

**Keywords:** viral zoonotic disease, (re-)emerging disease, land-use change, wildlife trade, livestock and domesticated animals, climate change, globalisation, geopolitics, sociology

## Abstract

Diseases that are transmitted from vertebrate animals to humans are referred to as zoonotic diseases. Although microbial agents such as bacteria and parasites are linked to zoonotic events, viruses account for a high percentage of zoonotic diseases that have emerged. Worryingly, the 21st century has seen a drastic increase in the emergence and re-emergence of viral zoonotic disease. Even though humans and animals have coexisted for millennia, anthropogenic factors have severely increased interactions between the two populations, thereby increasing the risk of disease spill-over. While drivers such as climate shifts, land exploitation and wildlife trade can directly affect the (re-)emergence of viral zoonotic disease, globalisation, geopolitics and social perceptions can directly facilitate the spread of these (re-)emerging diseases. This opinion paper discusses the “intelligent” nature of viruses and their exploitation of the anthropogenic factors driving the (re-)emergence and spread of viral zoonotic disease in a modernised and connected world.

## 1. Introduction

Of the thousands of pathogens known to infect humans, >70% are zoonotic and viral in nature [1]. Diseases that are transmitted from vertebrate animals such as reptiles, mammals and birds to humans are defined as zoonoses [2]. These animals, which serve as reservoirs or amplifier hosts, facilitate the initial jump from animals to humans through direct or indirect interactions [3]. Although viral zoonoses are a recurring process, humans are commonly dead-end hosts to these pathogens [4]. On the other side of the spectrum, some viruses can acquire genetic mutations within the new human host to sustain human-to-human infection. In a much rarer event, certain animal viruses can adapt to their human hosts so effectively that new host-exclusivity is eventually acquired and subsequently sustained [5]. Since most viruses cannot coexist with humans, they are unable to sustain replication and are therefore readily shed from blood as well as the gastrointestinal, urogenital and respiratory tracts without causing much harm, if any at all. In fact, most acute viral zoonoses require constant reintroduction from non-human hosts to begin human-to-human transmission [6].

Regardless of the rarity, an upward trend in the emergence and re-emergence of viral zoonoses has been observed over the past few decades. Notwithstanding their ability to rapidly mutate, viruses and their potential zoonoses are largely triggered by human influence such as deforestation, farming, population and societal dynamics [7]. Adding to these are the anthropogenic disturbances of complex, yet intricate, biodiversity within the natural ecosystems [8]. Given the role that humans play in the (re-)emergence of viral zoonoses, are our behaviours, beliefs and quest for modernisation responsible for our own demise? This article explores the adaptability of viruses and the consequences of human action in the (re-)emergence and subsequent spread of viral zoonotic diseases.

## 2. Are Viruses “Smarter” Than We Think?

Since the dawn of time, viral infections have scourged humankind [9]. While the earliest documented viral-related pandemic dates to 165 CE [10,11], a recent discovery of an outbreak which seemingly infected 300 villagers in China was attributed to an unknown viral agent over 5000 years ago. Since then, several pandemics, epidemics or outbreaks were found linked to coronaviruses, Marburg viruses, Ebola viruses, variola viruses (i.e., smallpox), human immunodeficiency viruses (HIVs), Zika viruses, measles viruses and several types of influenza viruses (Figure 1) [9,12,13]. Certainly, viral zoonoses have a habit of appearing sporadically and unexpectedly. Could it be that viruses are evolving for sustained infection and transmission?

For example, although the incubation period for the rabies lyssavirus varies, symptoms usually develop 20–90 days after exposure. In late stages of disease progression, the virus induces extreme paranoia by infecting specific parts of the brain. Paranoia leads to aggression which subsequently leads to greater transmission since the likelihood of infected animals attacking and biting other animals or humans is also high. Taken further, infection with the rabies lyssavirus also results in throat spasms when liquids are ingested. Since the infected animals tend to avoid water, this conceivably increases the viral load in the animals’ saliva [14,15]. It is now evident that these novel transmission routes were evolutionarily selected by the rabies lyssavirus to avoid extinction [16]. Another example is the Ebola virus. Between 2013 and 2016, the largest outbreak of Ebola virus disease was seen in West Africa, where sequencing data revealed that the circulating variant had an amino acid substitution coding for the virus’s surface glycoprotein that increased fusion with the human cell receptor NPC1. It was concluded that the virulence and transmission characteristics gained by the altered glycoprotein may have influenced the severity of the disease [17]. A similar scenario was observed in SARS-CoV-2, the aetiological agent of COVID-19. In 2020, Korber et al. [18] discovered a recurrent amino acid substitution at position 614 in the spike protein of SARS-CoV-2. The authors found that the shift from D614 to G614 in these variants resulted in higher viral loads, suggesting a fitness advantage for infectivity but not disease severity [18]. Currently, the D614 wild-type is no longer present in any of the circulating SARS-CoV-2 strains [19]. Another interesting example is the poliovirus. Polioviruses, which are transmitted via the faecal–oral route, are the causative agent of poliomyelitis. Although vaccinations have brought the virus close to eradication, where vaccines contain the live-attenuated virus, particularly polio type 2, the virus can incidentally revert to its virulent form, thus causing vaccine-derived poliovirus outbreaks [20]. However, it is important to note that most vaccine-derived poliovirus cases occur in populations with diminished immunisation coverage [21]. Additionally, due to the rigidity of the poliovirus vaccine, limited pathways exist for vaccine-derived immune escape pressure [22].

These examples show that viruses make “intelligent” changes to adapt to their hosts. It is well known that most viruses slowly become attenuated over time due to an initial accumulation of selectively advantageous mutations. As a result, these mutations are not fixed within the progeny populations [23]. An example of this was the 1889 flu pandemic which was linked to the human coronavirus OC43, a diverged variant of the bovine coronaviruses [24]. Since its initial peak, the virus has attenuated and currently causes the common cold in humans. Although attenuation is accompanied by negative markers such as diminished replication capacity, long-term coexistence with the host is a success as it allows the host to live longer and transmission to continue [25]. A more recent example is SARS-CoV-2 which developed a tropism toward the upper respiratory tract as opposed to lung tissue. The D614 mutation, as described earlier, produced the shift in tropism resulting in milder disease outcomes, shorter incubation periods and an affinity for greater transmission between people [26]. Therefore, pandemic-wise, is there a story repeating itself? What is evident is that, in each case, an interaction must occur between the virus and its newly acquired host. In Lewis Carroll’s *Through the Looking Glass*, the Red Queen said to a disgruntled Alice “*Now, here, you see, it takes all the running you can do, to keep in the same place*” [27]. This phrase, often associated with coevolutionary dynamics [28], simply implies that the more things change the more they stay the same. In this cycle of infections and reinfections, emergence and re-emergence, are we running fast enough? With human influence and the risk of (re-)emerging viral zoonoses at an all-time high, it is important that humans learn from these experiences and better prepare for future pandemics.

## 3. Human Impactful Drivers in the (Re-)Emergence of Viral Zoonoses

### 3.1. Land-Use Change and Its Intrinsic Role in the Species–Pathogen Biodiversity Interface

Land-use change is a term used to describe all human impactful effects on the use of land and its associated ecosystems at a global level [29]. It represents one of the most important drivers in the (re-)emergence of viral zoonoses [30]. Since deforestation, urbanisation, agriculture and livestock farming have large-scale impacts on the natural landscape, a domino effect is observed in pathogen and host species abundance, exposure rates as well as pathogen coevolution [31,32]. A study conducted by García-Peña et al. [33] found that in areas with high rodent species diversity, the expansion of croplands into pastures and forests increased the risk of zoonotic disease emergence through the circulation of several different types of pathogens such as hantaviruses. Moreover, a longitudinal study conducted by Plowright et al. [34] showed that pregnant and lactating flying fox bat species had a greater risk of Hendra virus infection, thus signifying the seasonal relevance of Hendra virus disease epidemics amongst the bats and likely zoonotic transmission to human populations. Interestingly, the authors also found that high viral seroprevalence was observed in flying foxes that showed nutritional stress when food sources were scarce, thereby inferring the negative impact of habitat loss on possible Hendra virus infection as well as transmission between different host populations [34].

Historically, two conflicting models for biodiversity-related zoonoses are theorised [35] as shown in Figure 2 below. In the first model, termed the amplification effect, diversified habitats are regarded as hotspots for new or emerging zoonotic pathogens since both pathogen and host species diversity is high [36,37]. The second model, known as the dilution effect, assumes that diversified habitats negatively correlate with the transmission of existing or re-emerging zoonotic pathogens [38,39]. Unlike the first model, the dilution effect has been subject to much debate and is highly controversial within the field of ecology, with some studies directly supporting it [39,40,41] while others still debate the factors that support it [42,43,44,45], particularly where zoonoses are concerned [35].

Elucidation of these models is complex and has often relied on the scarcity of selectively characterised information collected at the time of data analysis. For example, most research model assumptions are based on either (i) host–pathogen diversity; (ii) zoonotic host–pathogen diversity; or (iii) zoonotic host–pathogen abundance and diversity (Figure 2) for which data are collected and analysed within these selective niches. Nevertheless, each assumption relies on a common trend of opportunity, cross-species transmission and pathogenic establishment [35]. Importantly, multiple host species for the same zoonotic pathogen and the host’s ability to effectively transmit the pathogen must be considered in biodiversity-related zoonoses [5].

Regardless, infringements on natural biodiversity are a major contributor in the emergence of viral zoonoses [46]. For example, a systematic review by Tapia-Ramírez et al. [47] indicated that novel mammarenaviruses (i.e., viruses associated with viral haemorrhagic fever) were identified in 27 out of the 47 rodent reservoirs found in the Americas. The authors further stated that although virions could not be isolated from the 20 remaining rodent reservoirs, mammarenavirus antibodies were detectable in this subset. Dacheux et al. [48] performed a viral metagenomic study on insectivorous French bats in contact with humans. The study revealed that in addition to the known mammalian viral families found within the French bat species, several new mammalian viruses including gammaretroviruses and bornaviruses were identified. Interestingly, the authors also identified the first bat nairovirus (coined Ahun nairovirus) which was found to significantly diverge from all other nairoviruses identified to date [48]. While the former example demonstrates host species diversity, the latter demonstrates pathogen diversity and its potential for contributing to the emergence of viral zoonotic diseases.

Another example of species’ diversity was demonstrated by French et al. [49]. Here, the authors used meta-transcriptomic sequencing to investigate water-borne viral abundance and diversity in water samples collected from different anthropogenically affected sites along a single river in New Zealand. The authors found that 94% of the viral species identified were novel in nature and that 63 of those viral species may cause infection in water birds and fish. In addition, the study also found that viral species identified in water samples across the urban and farming areas were not present in the native forest sites [49]. This particular study may infer a potential zoonotic transmission route of virus to animal host to human host via direct animal contact or indirect contact with the animal’s excretions. Furthermore, the study also highlights that viral biodiversity is largely impacted by the environment. A further example of human encroachment on zoonotic disease emergence is the Ebola virus. Rulli et al. [50] showed that the Ebola virus spill-over events in West and Central Africa were linked to hotspots of habitat fragmentation suggesting that human interactions with wildlife were more common at these fragmented sites given their role in housing potential reservoir species.

Certainly, land-use change and its ecological impacts are complex and intricate processes that must be strategically balanced if civilisation is to conserve the natural ecosystem and prevent the (re-)emergence of zoonotic disease.

### 3.2. Wildlife Trade

The relationship between humans and nature is fragile and is often overlooked, particularly where wildlife trade is concerned. The business of trading wild or domesticated animals legally or not is a serious risk factor in the global spread of zoonotic disease [51]. It is estimated that more than one billion direct or indirect contacts between animals and humans occur annually [52]. To quantify the risk associated with global wildlife trade, Shivaprakash et al. [52] found that approximately 26.5% of mammals were natural carriers of ~75% of the known zoonotic viruses investigated in the study. The authors further suggest that, apart from rodents and bats, carnivores, primates and hoofed animals such as deer pose a serious zoonotic risk since 58% of the 228 known zoonotic viruses were collectively identified in this group of traded animals.

Illegal wildlife trade is a lucrative business, accruing USD 7–20 billion in revenue annually [53]. As crime syndicates have an integral role in illegal wildlife trade, it is difficult to control [54,55]. However, legal but poorly regulated trade such as that carried out in wet markets increases the risk of exposure and proximity of diverse species, thereby increasing the types of zoonotic pathogens potentially circulating within a single location [53].

Importantly, while most concern is placed on the role of bush meat in the (re-)emergence of zoonotic disease [55], the relocation of exotic animals for repopulation efforts, zoological facilities, domestication or eco-tourism should not be overlooked. These scenarios provide effective transmission routes for the introduction of novel and re-emerging zoonotic pathogens into the human host.

In 2003, a multistate outbreak of the Mpox virus resulted in 71 cases of human-to-human transmission after the importation of infected rodents by an exotic animal distributor in Texas [56]. Traceback investigations revealed that due to the proximity of the infected rodents to the distributors’ prairie dogs, animal-to-animal transmission had occurred. At this stage, the prairie dogs that were purchased by either the public or other animal distributors served as the secondary host for human-to-human transmission to occur [57]. More recently, a 53-year-old veterinary surgeon working at a research facility in Beijing that specialises in non-human primates was infected with monkey B virus, otherwise known as herpes B virus. Following the dissection of two monkeys, the individual presented with fever, nausea, vomiting and neurological symptoms, to which he eventually succumbed. Even though the monkey B virus has a 70–80% mortality rate in humans, its zoonotic spread is considered sporadic with the risk of secondary transmission appearing to be minimal [58]. Still, the risk of repetitive reintroductions may be what the monkey B virus requires to eventually gain a fitness advantage over humans.

Certainly, one of the most interesting demonstrations of a human-related zoonotic event was linked to a bacterial–viral coinfection seen in birds [59]. The authors found that a novel adenovirus (i.e., psittacine adenovirus HKU1) and bacterial *Chlamydophila psittaci* (*C. psittaci*) coinfection in mealy parrots led to a psittacosis outbreak in humans at an animal detention centre in Hong Kong. The authors further showed that the concentration of *C. psittaci* was higher when the viral load of adenovirus HKU1 was also high. The authors postulated that immune suppression caused by adenovirus HKU1 led to greater *C. psittaci* infections and higher bacterial loads, thus providing favourable conditions for zoonotic transmission to occur [59]. Indeed, the role of bacterial and viral coinfection in animals is an important risk factor in the (re-)emergence of zoonotic disease.

Finally, with adventure travel at an all-time high, the role of eco-tourism in zoonotic disease emergence cannot be ignored. Under these circumstances, the risk of contracting unknown pathogens is high since eco-tourism promotes activities such as safaris, extreme travel and adventure sports [60]. An example of such a scenario is the Balinese Hindu temple in Indonesia where macaques with previously characterised herpes B virus antibodies roam free. Given that the temple is a major tourist site, it has been suggested that a potential for zoonotic transmission exists due to close interactions between the tourists and indigenous macaques [61].

These examples indicate the extraordinary risk of choosing to house, work with or interact with exotic animals. On a grander scale, any dealings with wildlife and trade thereof pose an enormous risk to humanity. As it is unrealistic to imagine a world without such practices, it is therefore imperative that adequate institutional frameworks are developed and managed at an international level without disrupting funding and policies associated with biodiversity and conservation efforts.

### 3.3. Livestock and Domesticated Animals

Wildlife trade and exotic animals aside, a substantial number of zoonotic events are linked to human interaction with domesticated animals. In fact, a situation analysis conducted by the International Union for the Conservation of Nature found that 99% of ongoing zoonoses were linked to domesticated animals [62]. Due to its long-known human exclusivity, an underestimated example of a spill-over from domestic animals to humans is that of the measles virus (MeV) [63]. Given its close relation to the now eradicated cattle pathogen rinderpest morbillivirus, it is generally accepted that MeV emerged from cattle. A study conducted by Dux et al. [64] found that MeV likely arose circa 600 BCE, coinciding with large population numbers in several human settlements. Worryingly, there is still a large potential for other paramyxoviruses to emerge from livestock and result in a zoonotic event. Abdullah et al. [65] found that the peste de petits ruminant virus (PPRV) is only restricted from human cell entry by inadequate interactions with the human cell receptor SLAMFI. By using structural analyses, the authors further demonstrated that a single amino acid substitution in the haemagglutinin protein of PPRV favoured successful SLAMFI interactions and resulted in some escape from cross-protection and anti-MeV antibodies.

Another example is the zoonotic influenza viruses which frequently emerge from domesticated animals, particularly poultry and swine [66]. A study conducted by Mena et al. [67] found that the 2009 H1N1 influenza pandemic likely arose from infected swine in central Mexico and was perpetuated through global swine trade. Moreover, Graham et al. [68] indicated that the risk of H5N1 outbreaks in humans was significantly higher in commercial poultry farms, directly linking zoonotic disease prevalence and livestock production. Another important example was the Nipah virus outbreak that occurred in Malaysia in 1998. Chau et al. [69] found that deforestation efforts reduced the number of flowering and fruiting forest trees for foraging fruit bats, leading to encroachments in cultivated fruit orchards. As the orchards were proximal to piggeries, the Nipah virus was transmitted from the fruit bats to domesticated pigs to humans. Finally, high-mortality-rate viruses such as the Crimean Congo haemorrhagic fever (CCHF) virus may also be transmitted to humans through tick bites and contact with animal secretions, particularly in livestock farming areas [70].

### 3.4. Climate Change

Geoclimatic factors such as ocean and land temperature, wind patterns, severe weather as well as land characteristics have become important drivers in the transmission of infectious disease [71]. As the average global temperature continues to rise at an unprecedented rate [72], it is important to reconcile the effects of human-driven climate change on the incidence of disease [73]. Perhaps most notable is the link between climate change and vector-borne zoonoses [1]. Notwithstanding its role in altering natural ecosystems, the effect of climate change on the host, pathogen and vector can alter the (re-)emergence, geographical abundance as well as transmission dynamics of vector-borne diseases [74].

In a multiscenario, intercomparison modelling study, Colón-González et al. [75] showed that low greenhouse gas emissions corresponded with reduced transmission seasons and population risk for vector-borne diseases, dengue and malaria. A trait-based modelling study conducted by Shocket et al. [76] found that human cases of vector-borne West Nile virus peaked at 24 °C across the United States. The authors further suggest that global warming will likely shift disease dynamics, whereby the transmission of mosquito-borne viral disease will increase in cooler geographical areas as opposed to warmer locations, a finding supported by other research data [77,78,79]. However, Ryan et al. [80] also found that although a poleward expansion of mosquito-borne disease is likely, a shift in some lower-latitude locations to become hotter will result in unsuitable conditions for transmission, thereby shortening the season and resulting in no net increase in its overall spread. Another example is the impact of drought and the El Niño–Southern Oscillation effect on the transmission of mosquito-borne Rift Valley fever virus (RVFV), whereby unusual rainfall patterns lead to an abundance of vectors, thereby increasing infection in animals and humans [81,82,83]. A 5-year study on the post-epizoonotic period of RVFV transmission in the province of Free State of South Africa found that from 2015–2016 high surface temperatures coupled with severe drought and reduced vegetation resulted in unfavourable conditions for the breeding of virus’ mosquito vectors. However, the return of higher than normal rainfall during the 2017–2018 agricultural season resulted in a localised RVFV outbreak [84]. Further to these examples, several studies have linked the incidence of CCHFV to climate variables such as precipitation, temperature and humidity [85]. As ticks prefer dry and warm conditions that coincide with rising global temperatures, this may also lead to an increase in CCHFV vector populations and its spread. Changing climatic conditions may also allow for migratory bird populations carrying CCHFV to infect naïve livestock [85,86]. Therefore, a potential increase in CCHFV infections in livestock within chronically endemic areas may also lead to a greater likelihood of CCHFV in non-endemic locations due to international trade [70].

Vector-borne viral diseases aside, climate change can affect other mammalian spill-over events. Beyer et al. [87] found that regions around Central Africa, South and Central America, as well as a larger cluster within the province of Yunnan in China and adjacent Laos and Myanmar experienced an increase in bat biodiversity richness due to global greenhouse gas emissions, potentially increasing the risk of zoonotic spill-over events. Additionally, by using field surveillance data collected over a 54-year cycle from central China, Tian et al. [88] showed that temperature and rainfall were key factors in hantavirus transmission and reproduction of their rodent hosts. More recently, Ferro et al. [89] and Douglas et al. [90] supported these findings, showing direct relationships between temperature, rainfall, rodent host dynamics and hantaviral disease emergence in Latin America and the Caribbean. Furthermore, statistical modelling of Ebola virus zoonotic events over three decades in sub-Saharan Africa revealed that spill-over intensity is greatest between transitions of wet and dry periods and when human population numbers are either considerably high (1000/km^2^) or considerably low (<100/km^2^) [91]. Finally, climate projections made by Rupasinghe et al. [71] revealed that in addition to expanding host habitats, the increase in the intensity of water-borne, vector-borne, rodent-borne, air-borne and food-borne zoonotic events in the future will likely accelerate.

Although climate predictions are challenging, these examples highlight the importance of climatic factors in the (re-)emergence of viral zoonotic disease. Many gaps exist in our knowledge, but if we want to better plan and prevent the next pandemic, more studies around climate and its relation to host–viral ecology are urgently required.

## 4. Human Impactful Drivers Related to the Spread of (Re-)Emerging Viral Zoonotic Disease

### 4.1. Globalisation

Globalisation theory refers to interrelated societal engagements that extend beyond the boundaries of geographical and cultural differences. Immersed within globalisation is the concept of modernisation which refers to societal or geographical development through modern practices [92,93]. Although globalisation is meant to create a non-separatist culture towards global benefit, it may simultaneously promote disease through tourism, transportation, migration as well as international learning and trade [94]. While the association between globalisation and the spread of disease is highly underestimated, it is not a new occurrence. In fact, one of the earliest documented cross-border epidemics was the Athenian plague [95]. Occurring in 430 BCE, the disease, which was attributed to smallpox or typhus, was believed to have originated in Ethiopia and spread to Greece by boats containing grains [96]. More recently and from a viral zoonotic standpoint, Giorgio et al. [97] postulated that transmission of HIV was facilitated by globalisation in Africa. Specifically, the collapse of colonialism, international trade efforts, socio-political reform and cultural promiscuity were speculated to provide the necessary conditions for HIV to perfect itself for human exclusivity and transmission [98]. Another example is the 2014 Ebola virus epidemic. While the epidemic was initially linked to the consumption of bush meat, the international risk it posed was attributed to air travel between cities around the globe [99]. Perhaps one of the most notable examples is the recent COVID-19 pandemic. While the emergence of SARS-CoV-2 was thought to have originated in Wuhan, China, retrospectively, its containment may have been near impossible at the time. Notwithstanding the delayed health response, SARS-CoV-2 was rapidly and efficiently spread across the globe due to the ease of intercontinental travel and trade [100,101].

Furthermore, while the relocation of diseased vectors due to air travel is highly improbable, air travel itself can facilitate contact between an infected person and an invasive or native vector population, thus allowing for local transmission to occur [102]. For example, the origin and timing of the 2015–2016 Zika virus outbreak in Brazil are politically sensitive issues that have remained the subject of scientific disparity [103]. However, a phylogenetic assessment conducted by Zanluca et al. [104] found that viral sequences isolated from Brazil in March 2015 belonged to the Asian clade which had been previously circulating in the Pacific Islands. Similarly, Campos et al. [105] supported these findings, showing a 99% sequence identity between the isolates from Brazil and that of French Polynesia. Further to these, a mathematical study conducted by Massad et al. [103] revealed that based on the viral replication rate, force of infection in French Polynesia and volume of travel, the Zika virus responsible for the epidemic was likely exported from French Polynesia to Brazil. Influenza is another example of a virus that can be transmitted via air travel. Belderoc et al. [106] found that travellers to subtropical regions frequently contract influenza viruses due to its continuous circulation within these geographical locations. The authors further suggest that given the viral incubation period and high travel volume, travellers become vectors who can perpetuate the spread of influenza globally. Through contact tracing, Kim et al. [107] showed that amongst two clinically diagnosed individuals who were infected with influenza A and travelled on the same flight to Seoul from Los Angeles, one patient was shown to have acquired the virus in-flight.

It is evident that transnational infectious disease is not a new concept as echoes of past diseases still reverberate today. However, what is significant are the ever-increasing scale of globalisation and the microbial traffic thereof. While months to even years were required for transnational endeavours in the past, today cross-border activities require no more than a few hours to days. It is now more important than ever to maintain a balance between positive global dynamism and the factors contributing towards the (re-)emergence and transmission of viral zoonotic disease.

### 4.2. Geopolitics

Geopolitics represents a silent but important challenge in the control of disease [108]. In essence, geopolitics is a term used to describe the projection of power within a political and geographical landscape. As infectious diseases pose both economic and social threats, they have the potential to induce negative geopolitical effects [109]. For example, when human-to-human transmission of the H5N1 “bird flu” virus was rapidly occurring across Asia and it presented a 40% mortality rate, a vaccine research and development programme was developed to prepare for a possible global pandemic [101]. In doing so, viral samples isolated from infected humans were subsequently shared with laboratories across the globe. Indonesia, which had the highest number of human H5N1 cases at the time, implemented a “viral sovereignty”, where the samples were declared the property of the state, thus stopping sample sharing with other countries worldwide. This political doctrine implemented by the Indonesian government arose from an uncertainty of fairness regarding access to future biomedical interventions and benefits. Although this act was strongly condemned at the time, viral sovereignty remains today [101].

Another example of a geopolitical-related disease event was the Ebola virus outbreak that began in 2018 in the Democratic Republic of Congo. Despite the availability of therapeutic interventions, the health care response was afflicted by civil unrest and militant groups targeting medical workers [110]. Furthermore, delayed responses to previous Ebola virus epidemics in West Africa by global leadership meant that poorly developed countries with minimal to no resources had to respond to epidemics they were not equipped to handle [111]. From 1900–1902, the Boer War in South Africa saw rural farming families of Dutch heritage being placed in concentration camps by the British Army. While these camps were established for military purposes, they would eventually become disaster sites [112]. Confinement led to widespread transmission of measles as well as other acute respiratory infections [113]. Similarly, during the 1917–1918 World War, US military recruits placed in overcrowded mobilisation camps saw large-scale measles outbreaks in the mobilisation camps and in ships headed for Europe. Evidently, investigations into measles would later become important for the military’s response to the 1918 influenza pandemic [114], which infected one-third of the world’s population at the time [115] and resulted in approximately 50–100 million deaths [116]. Although the origin of the pandemic H1N1 virus is unknown [115], what is known is that the war played an integral role in viral seeding, providing an efficient mechanism for passage and mutation [116,117]. These examples represent a small fraction of the many geopolitical events known to have influenced the spread of (re-)emerging pathogens in the past.

### 4.3. Social Perceptions

Science is constantly changing. What is today may not be tomorrow. It is especially important to understand that when a scientific concept, finding or characteristic has changed, it is a direct consequence of having discovered more and not a matter of mistruth. Science endeavours to discover the unknown, but with every question answered more questions arise. Perhaps a more insightful way to think about it is that the act of science is trying to answer questions we have not yet asked.

Unfortunately, predictions of potentially emerging or re-emerging viral zoonotic diseases and their epidemiology are never completely accurate. Microbes, particularly viruses, are in a constant state of flux. Their ability to acquire adaptive genetic mutations in short periods of time is unprecedented. As a result, predictions rely on historical data that resemble the disease of interest [118]. Unfortunately, this does not always translate well when a new viral disease emerges. Perhaps one of the biggest challenges is maintaining the dissemination of information to the public and the media. While science communication by journalists can be extremely beneficial, one ambiguous statement can create disarray and havoc [119]. However, news or information sharing no longer relies solely on journalism. Rather, information may be easily shared across the globe via social networking, blogs, podcasts or any other platform linked to the internet by unqualified persons.

The early stages of the COVID-19 pandemic saw a slew of misinformation and conspiracy theories rapidly spreading despite open access to scientific research data. This led to distrust by the public at large [120]. A retrospective study conducted by Islam et al. [121] found that from January to April of 2020, 2311 cases of stigma, rumours and conspiracy theories were recorded in 87 countries across 25 languages. Reports were linked to causalities, illness, mortality, transmission, control, as well as treatment and cure. Worse still, misinformation led to 5876 hospitalisations, 60 cases of blindness and 800 deaths in 2020 [121]. Furthermore, preventative practices such as social distancing and mask wearing were challenged to the detriment of those at a higher risk [122]. Al-Ramahi and others [123] used machine learning techniques to analyse the link between negative attitudes to mask wearing and the number of new COVID-19 infections over 51,170 English tweets collected from January to October 2020. The authors found that negative tweets were strongly correlated with the number of new infections, where an increasing number of negative tweets preceded an increase in the number of new infections by nine days.

Another example of social perceptions as well as knowledge barriers was the Ebola virus disease outbreak in West Africa. Using data collected from 800 respondents in Ghana, Tenkorang [124] showed that social perceptions and inadequate knowledge on the Ebola virus led to unsafe burial practices that involved touching of the deceased. In Sierra Leone, Yamanis et al. [125] found that multiple respondents were sceptical of the validity on the Ebola virus test, with some persons only choosing to test after an individual had already died. Additionally, when presented with fever some individuals self-medicated, visited local health clinics or deferred medical attention altogether. This was born out of a need to either first confirm that the fever was due to Ebola sickness or out fear that they would die after treatment. Most of this stemmed from distrust of the government [125], and these practices likely increased the transmission of the virus from person to person, thereby prolonging the epidemic.

Perhaps one of the most impactful events was the Duesberg phenomenon which saw a Berkley virologist discredit scientific evidence that HIV was the aetiological agent of AIDS [126,127]. Although his claims were strongly rebutted by the greater scientific community, his voice reached many HIV denialists. Even in South Africa, where the prevalence of HIV is the highest in the world, the role of HIV in the development of AIDS was questioned by former president Thabo Mbeki at the beginning of the national antiretroviral rollout. Across the globe, medical practitioners and scientists were petitioning for the reinstatement of antiretrovirals and affirming the fact that AIDS was caused by HIV. Although the ban was eventually overturned due to international pressure, it was estimated that more than 330,000 people had died and at least 35,000 babies acquired HIV at birth [128].

## 5. Conclusions

As humans do not have acquired immunity against many of the emerging viral zoonotic diseases or have lost herd immunity against those that are re-emerging, more attention must be given to zoonotic detection, prevention and response. Since many of the known anthropogenic factors found contributing towards the (re-)emergence or spread of viral zoonoses are likely interconnected, such initiatives may only be accomplished on a multisectoral level. Therefore, partnerships such as One Health must remain relevant. Although the concept of One Health is not new, it has been reignited due to rapidly changing interactions between animals, people, plants and the environment. Therefore, to ensure the success of One Health and future global governance, the competitive interests of both public and private sectors must be dispelled. Additionally, more attention must be paid to the relationship between negative social perceptions and qualified scientific data, since social factors seem to represent some of the biggest and most uncontrollable factors in the transmission of disease. By closing the gaps between resource and land governance, conservation, sociology, disease ecology and geopolitics, we may be able to prevent future pandemics from occurring. The examples presented in this paper have not only shone a light on the absurdity of our actions but have also highlighted how our ignorance towards the microscopic world can lead to the (re-)emergence and spread of devastating diseases. If anything, COVID-19 taught us a valuable lesson and may have hopefully opened a way for change. After all, “*Those who cannot remember the past are condemned to repeat it*”—*George Santayana*.

## Figures and Tables

**Figure 1 viruses-15-01638-f001:**
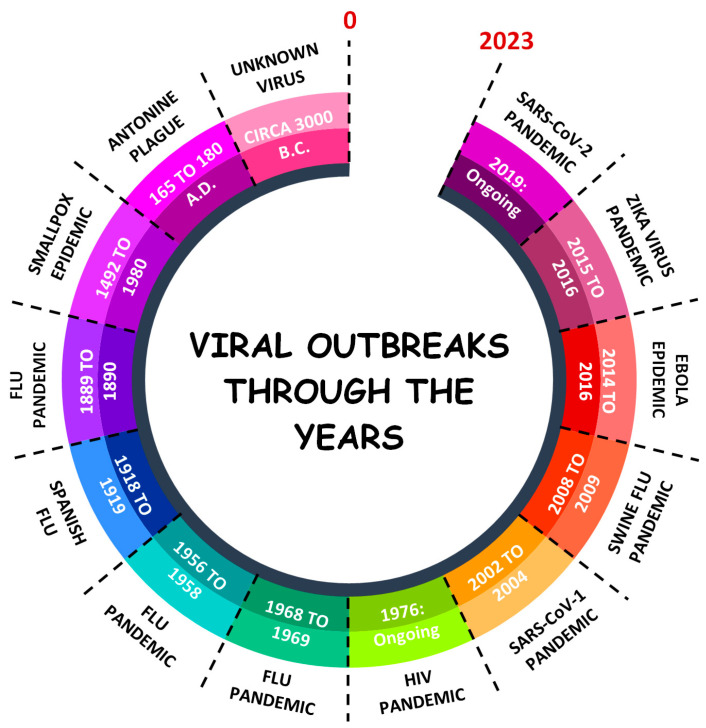
Timeline of history’s most notable viral pandemics and epidemics. Most major pandemics were attributed to mutated influenza viruses (H1N1, H2N2, H3N2) that were thought to have originated in animal reservoirs and which subsequently spread to humans. The 2003 SARS-CoV-1 pandemic is regarded as the first pandemic of the 21st century and, similar to SARS-CoV-2, likely emerged from bats. Note 1: Although the agent responsible for the outbreak 5000 years ago is not known, scientists hypothesise that due to the rapid mortality and transmission rate, the disease was likely caused by the measles virus. Note 2: Based on descriptions by Greek physician Galen, the Antonine plague may be attributed to smallpox.

**Figure 2 viruses-15-01638-f002:**
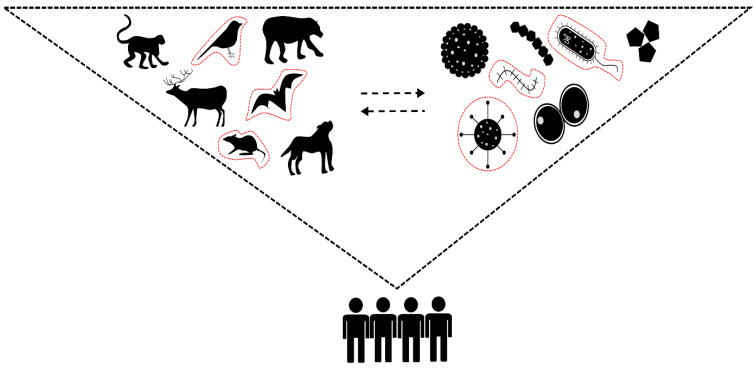
Alternative biodiversity models linking host and pathogen dynamics in the (re-)emergence of zoonotic disease. (1) In both the amplification and dilution models, total host diversity assumes a spectrum of circulating microbes that have the ability to jump the species barrier; (2) the zoonotic host species diversity model (animals circled in red) assumes that potential host species are more likely to harbour zoonotic pathogens (microbes circled in red); (3) the zoonotic pathogen–host species diversity and abundance model assumes that both host species diversity and the prevalence of pathogens determine the potential for zoonotic disease emergence (adapted from [35]).

## Data Availability

Not applicable.

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
