# Peer review of "The (Re-)Emergence and Spread of Viral Zoonotic Disease: A Perfect Storm of Human Ingenuity and Stupidity"

_viruses, 2023, doi:10.3390/v15081638_

Round 1

Reviewer 1 Report

The (re)emergence and spread of viral zoonotic diseases:A perfect storm of human ingenuity and stupidity land use and biodiversity. The authors first introduce the topic of emergence and re-emergence of zoonotic diseases and then focus on viral zoonotic diseases. The authors relate emergence of viral zoonotic diseases to the following: wildlife trade, climate change, and human drivers such as globalization, geopolitics, and social perceptions using several examples of how each affects zoonotic disease transmission, historically or currently. These areas are useful and appropriate to consider. The authors conclude that we need to address these in order to prevent or at least better deal with emergent zoonotic diseases in the future. The manuscript is well written and will likely have average to above average interest for readers.

Comments:

Abstract: Sentence 1 - "referred to as zoonotic disease." Should this be plural?

Abstract: Sentence 2- "... that have emergence." Should this be "emerged?"

Intro End first paragraph "...excreted from the gastrointestinal tract..." Why not include respiratory and urogenital as well?

2. "Are viruses smarter..." To indicate that you are taking 'poetic license' here why not place "smarter" in quotes like the case in other statements about the intelligence of viruses?

2. Second Paragraph- Is this rabies description a good example of viral evolution? Have we evidence that the clinical course of rabies has changed over the centuries in either man or animals? The other examples in this section (2) are really strong.

2. Ending statement- "With human influence and the risk of (re)emerging .... better prepare for future pandemics" Agree totally.

3. Land use change and biodiversity. Not clear if the intent is "Land use change" on its own and "biodiversity" on its own or "Land use change AND biodiversity" Linkage of these concepts in this section is not strong until near the end of the section.  Changes to the bulk of this section's text is not recommended. Just consider editing the 3.1 header?

3.2 Wildlife trade 4th Paragraph- "It was reported that upon dissecting two monkeys..." Recommend changing this to "following the dissection of two monkeys..."  From China CDC Weekly: This case of BV occurred in a veterinary surgeon (53 years old, male) who worked in an institute specialized in nonhuman primate breeding and experimental research in Beijing. He dissected two dead monkeys on March 4 and 6, 2021 and experienced nausea and vomiting followed by fever with neurological symptoms one month later.

3.2 Wildlife trade 5th Paragraph- "...most interesting demonstration of zoonoses was linked..." It was not immediately obvious to me that this meant transmission to humans. Could have been 'reverse zoonoses'. Suggest adding 'human' in the first line as this is a good example.  From the cited publication: "This is the first documentation of an adenovirus-C. psittaci co-infection in an avian species that was associated with a human outbreak of psittacosis. Viral-bacterial co-infection often increases disease severity in both humans and animals. The role of viral-bacterial co-infection in animal-to-human transmission of infectious agents has not received sufficient attention and should be emphasized in the investigation of disease outbreaks in human and animals.

3.3 Climate change 1st Paragraph- "... as land characteristics has become..." Suggest "have become"

 3.3 Climate change 2nd Paragraph- "...viral disease will increase in cooler geographic areas..." Suggest "in currently cooler geographic..." Also, recommend mentioning that the increase in cooler areas may be balanced by a decrease in currently warmer areas resulting in a no net overall increase in transmission. See the Ryan 2015 pub.

References:

From the Viruses For Authors page:

  • Journal Articles:
    1. Author 1, A.B.; Author 2, C.D. Title of the article. Abbreviated Journal Name YearVolume, page range.
  •  
  •  
  •  

 All: year follows Journal name.

All volume is italicized 

All need a page range if one exists. 

Reviewer 2 Report

Major

This reviewer feels it is necessary for the authors to include more detail about the emergence of zoonotic disease from livestock and domesticated animals. A recent report by International Union for Conservation of Nature (IUCN) determined that contact with domesticated animals accounts for 99% all recurring zoonoses, while the wildlife trade is only linked to 1% (1). Based on this information this reviewer feels, contact with livestock and domesticated animals warrants a whole subsection in section 3 (Human impactful drivers in the (re)emergence of viral zoonoses).

To highlight this point, I have listed several examples of human contact with domesticated animals leading zoonoses that could be included.

1.       The emergence of the Measles Virus from spill-over events with a bovine paramyxovirus. It is generally accepted now that measles emerged due a spill over event with a bovine virus (2). One study dates the emergence of measles to coincide with the rise of big cities (3). Furthermore, there is still potential for other livestock paramyxoviruses to emerge as zoonotic pathogens. Peste des petits ruminants virus (PPRV) is a close relative of measles that infects goats and sheep. Studies have shown that PPRV requires just 2 point mutations in its glycoprotein to enable it to use the human version of its receptor (4).

2.       Zoonotic flu viruses frequently evolve from domesticated swine and poultry viruses. Evidence suggests that H5N1 outbreaks are becoming more frequent due to intensive farming (5).  The 2009 outbreak of H1N1 originated on a Pig farm (8).

3.       The 1998 Malaysia outbreak of Nipah virus outbreaks were caused by pig to human transmission. The 1998 Malaysia outbreak of Nipah virus, was caused by spread of the virus from pigs to farm and abattoir workers (7). It is thought that deforestation and climate change also played a role. The primary host of Nipah virus is the fruit bat, deforestation and climate change led fruit bats to roost close to pig farms in Malaysia. The fruit bats then transmitted Nipah virus to pigs, which only develop mild symptoms. This led to an amplification of the virus in pig who transmitted it to humans, where it causes a much more serve disease.

4.       Livestock to human transmission is the main causes of CCHFV in humans. CCHFV causes a haemorrhagic disease in humans with mortality rates of over 30%. The disease is usual transmitted by ticks to wildlife and livestock animals such as cows and sheep. In these hosts it generally causes a mild or asymptomatic infection. Outbreaks of CCHFV are frequently linked to people who come in contact with infected animals (8).

Minor points.

In addition to this I would also recommend making the following minor amendments, additions or corrections. On some points I have suggested further examples of the phenomenon that the reviewers have already highlighted. The reviewer feels including these additional examples would help reinforce the authors argument.

1.       Page 1. “Since most viruses cannot co-exist with humans, they cannot sustain replication and are therefore readily excreted from the gastrointestinal tract with-out much harm, if any at all. In fact, most acute viral zoonoses require constant re-intro-duction from its non-human host to begin human-to-human transmission [6].” The way this sentence reads at present implies all viruses infect via the gastrointestinal route. It would be useful to reword it to take account of respiratory and blood borne pathogens. 

2.       Page 1. “Notwithstanding its ability to rapidly mutate, viruses and their potential zoonoses are largely triggered by human influence such as population and societal dynamics”. It would be useful to briefly introduce one or two human influences here. E.g. deforestation and farming.

3.       Page 3. “The authors found that the shift from D614 to G614 in these variants resulted in higher viral loads, suggesting a fitness advantage for infectivity but not disease severity. Presently, the D614G mutation has been expectedly found in all major strains of SARS-CoV-2 [18].” From my understanding the D614 is no longer present in any circulating strains. It would be useful to mention this, as it would reinforces your point further.

4.       Page 3. “The virus can incidentally revert to its virulent form thus causing vaccine-derived poliovirus outbreaks [19].” It is possible that readers could misunderstand this point, and assume that vaccine derived polio is associated with vaccine escape mutations. It is important to highlight this is not the case and to state there are no vaccine escape mutations and that vaccine derived polio outbreaks only occurs in population with low vaccines coverage.

5.       Page 3. Is it known if when OC43 gained the ability to infect humans did it also loose the ability to infect cows? If there is any information on this please include it.

6.       Page 3. “A study conducted by García-Peña et al. [30] found that in areas with high rodent species diversity, the expansion of croplands into pastures and forests increased the risk of zoonotic disease emergence through the circulation of several different types of pathogens.”  Could you include an example of one of these zoonotic rodent viruses.

7.       Another example useful example of reemergence of a zoonotic virus due to human encroachment on biodiversity that could be included is Ebola.

8.       Page 6. A further example that could be included of a zoonotic virus that is affected by climate change is CCHFV (8).

9.       Page 7. Air travel has also been linked to the spread of Influenza (10, 11).

10.   Page 9. “humans have little to no existing immunity against many of the emerging viral zoonotic diseases”. Please reword this to highlight that humans do not have acquired immunity towards zoonotic diseases. Parts of the innate immune system can still act as protective barrier to some zoonotic diseases.

References

1.       Kock, R. and Caceres-Escobar, H., Situat. Anal. roles risks Wildl. Emerg. Hum. Infect. Dis. 2022

2.       Wolfe ND, Dunavan CP, Diamond J, Origins of major human infectious diseases. Nature 447, 279–283 (2007)

3.       Düx A, Lequime S, Patrono LV, Vrancken B, Boral S, Gogarten JF, Hilbig A, Horst D, Merkel K, Prepoint B, Santibanez S, Schlotterbeck J, Suchard MA, Ulrich M, Widulin N, Mankertz A, Leendertz FH, Harper K, Schnalke T, Lemey P, Calvignac-Spencer S. Measles virus and rinderpest virus divergence dated to the sixth century BCE. Science. 2020 Jun 19;368(6497):1367-1370. doi: 10.1126/science.aba9411. PMID: 32554594; PMCID: PMC7713999.

4.       Abdullah N, Kelly JT, Graham SC, Birch J, Gonçalves-Carneiro D, Mitchell T, Thompson RN, Lythgoe KA, Logan N, Hosie MJ, Bavro VN, Willett BJ, Heaton MP, Bailey D. Structure-Guided Identification of a Nonhuman Morbillivirus with Zoonotic Potential. J Virol. 2018 Nov 12;92(23):e01248-18. doi: 10.1128/JVI.01248-18. PMID: 30232185; PMCID: PMC6232486.

5.       Graham JP, Leibler JH, Price LB, Otte JM, Pfeiffer DU, Tiensin T, Silbergeld EK. The animal-human interface and infectious disease in industrial food animal production: rethinking biosecurity and biocontainment. Public Health Rep. 2008 May-Jun;123(3):282-99. doi: 10.1177/003335490812300309. PMID: 19006971; PMCID: PMC2289982.

6.       Mena I, Nelson MI, Quezada-Monroy F, Dutta J, Cortes-Fernández R, Lara-Puente JH, Castro-Peralta F, Cunha LF, Trovão NS, Lozano-Dubernard B, Rambaut A, van Bakel H, García-Sastre A. Origins of the 2009 H1N1 influenza pandemic in swine in Mexico. Elife. 2016 Jun 28;5:e16777. doi: 10.7554/eLife.16777. PMID: 27350259; PMCID: PMC4957980.

7.       Chua KB, Chua BH, Wang CW. Anthropogenic deforestation, El Nino and the emergence of Nipah virus in Malaysia. Malays J Pathol. 2002;24(1):15–21

8.       Spengler JR, Estrada-Peña A, Garrison AR, Schmaljohn C, Spiropoulou CF, Bergeron É, Bente DA. A chronological review of experimental infection studies of the role of wild animals and livestock in the maintenance and transmission of Crimean-Congo hemorrhagic fever virus. Antiviral Res. 2016 Nov;135:31-47. doi: 10.1016/j.antiviral.2016.09.013. Epub 2016 Oct 3. PMID: 27713073; PMCID: PMC5102700.

9.       Estrada-Peña A, Zatansever Z, Gargili A, Aktas M, Uzun R, Ergonul O, Jongejan F. Modeling the spatial distribution of crimean-congo hemorrhagic fever outbreaks in Turkey. Vector Borne Zoonotic Dis. 2007 Winter;7(4):667-78. doi: 10.1089/vbz.2007.0134. PMID: 18047397.

10.    Belderok SM, Rimmelzwaan GF, van den Hoek A, Sonder GJ. Effect of travel on influenza epidemiology. Emerg Infect Dis. 2013 Jun;19(6):925-31. doi: 10.3201/eid1906.111864. PMID: 23735636; PMCID: PMC3713810.

11.    Hollingsworth TD, Ferguson NM, Anderson RM. Frequent travelers and rate of spread of epidemics. Emerg Infect Dis. 2007 Sep;13(9):1288-94. doi: 10.3201/eid1309.070081. PMID: 18252097; PMCID: PMC2857283.

Reviewer 3 Report

The opinion article entitled "The (re)emergence and spread of viral zoonotic disease: A per-fect storm of human ingenuity and stupidity", is a well written piece outlining past and current threats of emerging zoonotic disease. The articles is of interest and does a nice job outlining risk factors that contribute to transmission providing points of support.

There are a few minor comments to consider,

-The title outlines human ingenuity ad stupidity, The are points touched on throughout the article to highlight the fact that overall humans should be aware on consious of past events to better prepare however it might help to provide a more forward statement in the conclusion to accentuate the title. 

Within the abstract: Although humans are contributing to climate change, it may be more correct to alter this statement. The line “While human impactful drivers such as …..” should be changed to “Drivers such as climate shifts, land exploitation and wildlife trade”

In section 3.1: - Figure 2 caption could be modified to be clearer, specifically - “zoonotic pathogens (microbes circled in red) and zoonotic pathogen and host diversity as well as abundance assumes that both species diversity and the prevalence of pathogens determines the potential for zoonotic disease emergence.”

In section 3.2: 

-  Typo in the third line “animals, legally or no is a serious” should read not I believe.

-          In this section, another important factor to consider is the increase in eco-tourism. This is a serious threat to introduction of zoonotic disease where humans are crossing into biodiverse areas and should be considered as a significant means for zoonotic transmission.

 In the Climate change section:

-          CCHFV could be considered aa significant threats by the WHO and is rapidly expanding. Tick populations are being transported through migration and surviving due to warmer climates. This is evident especially in Europe or places like Iran with significant increases in cases. 

Round 2

Reviewer 2 Report

I am satisfied with all amendments made and look forward to seeing the final publication.